# Effects of Online Learning Readiness and Online Self-Regulated English Learning on Satisfaction with Online English Learning Experience During the COVID-19 Pandemic

**DOI:** 10.3390/bs15010093

**Published:** 2025-01-20

**Authors:** Sarah W. S. Ip, Wai-Ming To

**Affiliations:** Faculty of Business, Macao Polytechnic University, Macao SAR, China; wsip@mpu.edu.mo

**Keywords:** online learning readiness, online self-regulated English learning, satisfaction with the online English learning experience, perceived English self-efficacy, Chinese

## Abstract

The COVID-19 pandemic has caused major changes in pedagogical practices worldwide. As COVID-19 cases increased, universities had to move their teaching online, requiring both instructors and students to engage through online learning platforms. This study explored the effects of students’ online learning readiness and online self-regulated English learning on their satisfaction with the online English learning experience. Additionally, it investigated whether and how online learning readiness and online self-regulated English learning influenced students’ perceived English self-efficacy which could in turn influence their satisfaction with the online English learning experience. Data were collected from 163 university students in Macau, China. The results of partial least squares structural equation modeling showed that students’ online learning readiness significantly influenced their satisfaction with the online English learning experience directly and indirectly through online self-regulated English learning. Furthermore, online learning readiness affected students’ perceived English self-efficacy. However, online self-regulated English learning did not significantly impact students’ perceived English self-efficacy, and there was no significant link between students’ perceived English self-efficacy and their satisfaction with the online English learning experience. The implications of this study are given.

## 1. Introduction

English is recognized as the most popular business language globally. It has been widely taught as a second or foreign language across primary, secondary, and higher education institutions in various regions, particularly in the Asia–Pacific area ([38]). Studies on second or foreign language acquisition indicate that various factors, such as instructors’ teaching approaches, students’ motivational factors, and students’ self-regulated learning strategies, can influence students’ satisfaction with their English learning experience and perceived English self-efficacy, encompassing skills in listening, speaking, reading, and writing ([4]; [19]; [34]; [48]). Moreover, research highlights that self-regulated learning and self-efficacy are crucial precursors to students’ satisfaction with their learning experience and learning outcomes ([4]; [10], [9]; [36]). Additionally, the integration of e-learning and online learning has become a fundamental aspect of higher education over the past two decades ([22]; [29]; [37]; [44]). Consequently, university instructors have started to adopt blended learning approaches, which combine traditional face-to-face classroom instruction with e-learning and online elements, such as Internet-based chat rooms, discussion forums, and online self-assessment tools for English language teaching ([7]; [41]; [43]). In e-learning and online environments, students’ self-regulated learning strategies are pivotal in influencing learning outcomes ([8], [9]), especially given the reduced control instructors have in these environments. Furthermore, online learning readiness has been identified as a precursor to online self-regulated learning ([26]).

Between 2020 and 2022, the COVID-19 pandemic forced universities to adopt online teaching and learning due to the surge in COVID-19 cases ([17]; [48]). However, many university instructors and students were not fully prepared to teach and learn in virtual classroom environments, leading to enormous stress for both groups ([50]). Research shows that online learning negatively affects communication between instructors and students and hinders interactions between instructors and students as well as interactions among students ([1]). Consequently, it is essential to investigate how online learning readiness influences students’ satisfaction with the online English learning experience, both directly and indirectly through online self-regulation. Furthermore, there is limited research examining the complex relationships between students’ online learning readiness, online self-regulation English learning, perceived English self-efficacy, and satisfaction with the online English learning experience. To fill these research gaps, this study seeks to answer the following questions: First, does students’ online learning readiness directly and indirectly impact their satisfaction with the online English learning experience through online self-regulated English learning? Second, how do students’ online learning readiness, online self-regulated English learning, and perceived English self-efficacy collectively influence their satisfaction with the online English learning experience? This study offers three key contributions. Firstly, it expands the existing literature on English education by highlighting online platforms as vital resources for English education, especially during and beyond the COVID-19 pandemic. The trend is likely to gain momentum as innovative tools like ChatGPT and other large language models become more accessible. Secondly, it contributes to the existing literature on online English learning by highlighting online learning readiness and online self-regulated English learning as crucial factors influencing perceived English self-efficacy and students’ satisfaction with the online English learning experience. Thirdly, this study posits and illustrates that most of the studied constructs such as online learning readiness, online self-regulated English learning, and perceived English self-efficacy should be characterized by second-order factor structures, and the underlying factors were measured quantitatively.

## 2. Literature Review and Hypothesis Development

Online learning has gained popularity since the beginning of the 21st century. This shift occurred as many higher education institutions began to make their distance learning program materials available online, enabling enrolled students to access resources such as PowerPoint presentations, videos, and course assignments via the Internet ([3]; [32]). Subsequently, in the early 2010s, prestigious universities like Stanford, MIT, Harvard, and UC Berkeley launched Massive Open Online Courses (MOOCs) ([40]). These MOOCs attracted participation from over 100,000 students globally, prompting universities and educators to recognize the potential of online learning to improve the quality of teaching and learning ([46]). In traditional classroom settings, some instructors have begun to incorporate online learning elements into their courses, utilizing interactive e-learning platforms to foster blended learning environments ([18]). While blended learning and online learning present several advantages over conventional face-to-face instruction—such as shifting from instructor-led to student-centered learning, encouraging student independence, and facilitating collaborative learning ([20]), these benefits may not be fully realized if students are not adequately prepared and proactive in managing their time and learning.

### 2.1. Online Learning Readiness, Online Self-Regulated English Learning, and Satisfaction with Online English Learning Experience

COVID-19 suddenly appeared and spread quickly in China and other regions. In the initial nine months, over 20 million COVID-19 cases were reported globally. By November 2022, the total number of cases worldwide surpassed 0.6 billion ([16]), with an approximate infection fatality rate of 1%. The COVID-19 pandemic necessitated a shift to online teaching and learning at universities, compelling both instructors and students to remain at home and engage through online platforms.

While online learning gained prominence during the pandemic, it was not a novel concept; numerous distance learning programs and higher education institutions in the U.S. had been utilizing online education for many years ([3]; [32]). In these online environments, students’ online learning readiness—defined as their beliefs regarding their technical and social competencies—significantly influences the process and outcomes of online learning ([26]; [45]). Specifically, [45] ([45]) explored the importance of online learning readiness and created an instrument to measure it. Using responses from 311 university students who participated in at least one online course, [45] ([45]) found that technical competencies were a key factor for students’ online learning readiness, followed by social competencies with instructors and social competencies with classmates. [26] ([26]) further assert that online learning readiness positively predicts students’ self-regulated learning. They also emphasize that students’ technical competencies are vital for a successful online learning experience, as those with high technical competencies are better equipped to implement effective self-regulated learning strategies.

Self-regulated learning refers to learners’ self-directive processes that enable them to transform mental abilities into an academic performance skill ([49]). Self-regulated learning processes encompass planning, self-monitoring, self-evaluation, self-reflection, and the application of effort strategies. Importantly, self-regulated learning is influenced by the specific context in which it occurs. [34] ([34]) reviewed students’ online self-regulation and self-efficacy in the context of learning English as a foreign language. Their findings indicate that effective online self-regulated English learning encompasses elements like goal setting, structuring the learning environment, managing time, and conducting self-evaluations. Furthermore, online self-regulated English learning can predict students’ perceived learning outcomes such as perceived English self-efficacy ([34]). Additionally, [30] ([30]) highlighted that online self-regulated English learning significantly predicted students’ satisfaction with their online English learning experience in Vietnam.

[42] ([42]) investigated how students’ perceptions and readiness impact their online learning performance and satisfaction. Their study, which involved 356 university students in Taiwan enrolled in a general education asynchronous online course, revealed that online learning readiness in the form of technical competencies had a notable effect on students’ satisfaction with the course. Moreover, a strong correlation was found between students’ online self-regulated learning and their satisfaction with online courses ([5]; [21]). Consequently, this study posits the following:

**H1.** 
*Students’ online learning readiness positively influences their satisfaction with the online English learning experience.*


**H2.** 
*Students’ online learning readiness positively influences their self-regulated English learning.*


**H3.** 
*Students’ self-regulated English learning positively influences their satisfaction with the online English learning experience.*


Based on the above hypotheses, this study proposes the following:

**H4.** 
*Self-regulated English learning mediates the relationship between online learning readiness and satisfaction with the online English learning experience.*


### 2.2. Perceived English Self-Efficacy

Students’ perceived English self-efficacy refers to their beliefs about how well they can successfully perform tasks in English based on their past experience ([34]). This self-efficacy can be categorized into four main areas: listening, speaking, reading, and writing. According to [34] ([34]), students’ online self-regulated English learning was a predictor of their perceived English self-efficacy. Similar conclusions were drawn by [19] ([19]). Building on the findings of [19] ([19]) and [34] ([34]), this study posits the following:

**H5.** 
*Students’ self-regulated English learning positively influences their perceived English self-efficacy.*


[24] ([24]) investigated the relationships between students’ authentic language learning, self-directed learning, collaborative learning, and perceived English self-efficacy within online settings during the COVID-19 pandemic. Analyzing data from 529 university students in China, [24] ([24]) found that authentic language learning had a significant impact on perceived English self-efficacy, both directly and indirectly through self-directed learning and collaborative learning. Notably, in Lian et al.’s (2021) research, the term “self-directed learning” is a synonym for self-regulated learning ([28]). In Vietnam, [30] ([30]) discovered a significant correlation between students’ perceived English self-efficacy and their satisfaction with the learning experience, while online learning readiness, particularly in the form of technical competencies, was associated with perceived English self-efficacy and satisfaction. Consequently, this study posits the following:

**H6.** 
*Students’ online learning readiness positively influences their perceived English self-efficacy.*


**H7.** 
*Students’ perceived English self-efficacy positively influences their satisfaction with the learning experience.*


Based on H1 and the above two hypotheses, this study proposes the following:

**H8.** 
*Student’s perceived English self-efficacy mediates the relationship between online learning readiness and satisfaction with the online English learning experience.*


Figure 1 shows the theoretical model of this study. It illustrates the complex relationships between online learning readiness, self-regulated English learning, perceived English self-efficacy, and satisfaction with the online English learning experience.

## 3. Method

### 3.1. Sample and Data Collection

Macao is the premier gaming center globally, attracting nearly 40 million visitors annually prior to the COVID-19 pandemic ([25]). Despite its small size, with a population of 0.68 million and an area of 33 km^2^, the city boasts 10 higher education institutions that offer 342 programs, catering to over 39,000 enrolled students. A significant portion of postgraduate students and some undergraduate students are from mainland China. English courses are mandatory for nearly all undergraduate students, as the majority of bachelor’s programs in Macau are conducted in English. This study involved undergraduate business students enrolled in English courses at a public university in Macau, who were invited to participate in a questionnaire survey conducted from November 2022 to March 2023, during and immediately following the COVID-19 pandemic. The research team collaborated with English instructors to distribute the questionnaires to 250 students during their classes. To ensure that this study possessed sufficient power to identify significant effects, a priori power analysis was performed utilizing G*Power 3.1 software (Heinrich-Heine-Universität Düsseldorf, Düsseldorf, Germany; https://www.psychologie.hhu.de/arbeitsgruppen/allgemeine-psychologie-und-arbeitspsychologie/gpower (assessed on 15 October 2022)). The analysis was predicated on the following parameters: effect size (*f*^2^) set at 0.15 (indicating a medium effect size); alpha level (*α*) of 0.05; power (1-*β*) of 0.80; and a total of three predictors. Upon inputting these values, the minimum sample size necessary to detect a statistically significant effect was determined to be 77. Participation in this study was entirely voluntary and anonymous. The questionnaire commenced with a brief overview of this study’s purpose. Students were then required to provide their consent before proceeding with the questionnaire. Assurance was given regarding the confidentiality of their responses, and participants had the option to withdraw from the survey at any point. Due to an outbreak of the COVID-19 omicron variant in Macao from 18 June 2022 to 2 August 2022 ([35]), universities mandated that instructors conducted their first few lectures online in August 2022. Face-to-face classes were only allowed to resume after 1 September 2022. When completing the questionnaire, all participants were asked to reflect on their online English learning experience in August 2022. Ultimately, 163 out of 250 students returned the completed questionnaires, yielding a response rate of 65.2%, which surpassed the minimum number sample of 77 as determined by G*Power. Among the 163 respondents, 78 were male and 85 were female. The majority of respondents (71) were under 20 years, followed by 52 students aged between 20 and 22 years. Additionally, 97 respondents were in their first year of study, and 110 were pursuing a management degree. Among the 163 respondents, 75 preferred face-to-face classes only, 60 preferred hybrid mode of learning, and 28 preferred online English learning only. Table 1 shows the demographic profile of the respondents.

### 3.2. Measures

The questionnaire was divided into two sections. The first section focused on assessing students’ online learning readiness, self-regulated English learning, perceived English self-efficacy, and satisfaction with the online English learning experience. The second section comprised five questions aimed at gathering demographic data, including gender, age, year of study, specialization, and preferred learning mode.

To ensure content validity, the measures and items were either adopted or adapted from existing literature. Online learning readiness reflects students’ beliefs about their own competencies in utilizing online technologies ([26]; [45]). This construct encompasses three primary factors: technical competencies (4 items), social competencies with instructors (4 items), and social competencies with classmates (3 items). The items related to online learning readiness were sourced from [45] ([45]). Self-regulated English learning pertains to students’ self-directed practices that students employ to enhance their online English learning effectively ([34]). This includes four factors totaling 11 items, goal setting (3 items), environmental structuring (3 items), time management (3 items), and self-evaluation (2 items), all of which were derived from [34] ([34]). Perceived English self-efficacy is defined by students’ beliefs on how well they can perform tasks in English ([34]), measured through 20 items that assess self-efficacy in listening (5 items), speaking (7 items), reading (3 items), and writing (5 items). These items were adapted from [34] ([34]). Lastly, satisfaction with the online English learning experience measures the extent to which students were satisfied with different aspects of online English learning. Five items were adapted from [48] ([48]). Responses were recorded on a 5-point Likert scale, where 1 indicated “strongly disagree” and 5 indicated “strongly agree”. The measures and items are shown in Table 2.

### 3.3. Data Analysis Approach

Data were entered into an Excel file, where means and standard deviations were calculated to evaluate the agreement levels with the Likert scale items. This analysis provided insights into respondents’ perceptions of their readiness for online learning, the degree to which they engaged in online self-regulated English learning, their self-efficacy in English across listening, speaking, reading, and writing, and their satisfaction with the online English learning experience and the associated variances. To explore the relationships among constructs and the loadings of indicators, partial least squares structural equation modeling (PLS-SEM) was utilized. PLS-SEM is particularly advantageous for small sample sizes and complex models, especially when predicting a specific outcome such as students’ satisfaction with their online English learning experience ([12]). The analysis was conducted using SmartPLS, which was obtained from [33] ([33]). SmartPLS generates outer loadings for all measurement items, along with Cronbach’s alpha values, composite reliabilities (CRs), and average variance extracted (AVE) values. These metrics were utilized to evaluate item and construct reliabilities, as well as convergent and discriminant validities ([12]; [33]). The standardized path coefficients for the inner model were determined to reveal the direction, magnitude, and significance of the structural relationships among the primary constructs, which include online learning readiness, online self-regulated English learning, perceived English self-efficacy, and satisfaction with the online English learning experience. Furthermore, the PROCESS Macro ([15]) was employed to validate the mediating roles of self-regulated English learning and perceived English self-efficacy in the relationship between online learning readiness and satisfaction with the online English learning experience.

## 4. Results

Table 2 presents the means and standard deviations for the measurement items. Among the 11 items assessing online learning readiness, the means varied from 3.05 to 3.79. Notably, the means for the four items related to technical competencies were significantly above the midpoint of the 5-point Likert scale, which is 3.0 (*p* < 0.001). For the 11 items concerning online self-regulated English learning, the means ranged from 2.78 to 3.47, with the means of the 3 items focused on environmental structuring also significantly exceeding the midpoint of 3.0 (*p* < 0.001). In the domain of perceived English self-efficacy, 20 items were evaluated, yielding means from 2.72 to 3.45. The lowest mean, 2.72, was associated with the “listening” item “I can understand English movies (without Chinese subtitles)”, followed by the “reading” item “I can read English newspapers” with a mean of 2.74. Conversely, the highest mean of 3.45 was recorded for the “writing” item “I can write English compositions assigned by my English instructor”, closely followed by a mean of 3.44 for the “speaking” item “I can introduce myself in English (without reading notes)”. Among the five items measuring satisfaction with the online English learning experience, the means ranged from 2.78 to 3.58. Specifically, the highest mean of 3.58 was associated with the item “online English learning is time-flexible”, followed by the mean of 3.48 for the item “online English learning allows more autonomy and I can learn English in my own way”. Additionally, Table 2 includes Cronbach’s alpha values, which ranged from 0.754 to 0.946, all exceeding the recommended threshold of 0.70 as per [13] ([13]). A series of t-tests and ANOVAs were conducted to analyze the impact of demographic variables such as gender, age group, and year of study on the measurement items. The results indicated that most items did not show significant differences, implying that demographic factors did not significantly influence the variables under investigation.

### 4.1. Outer Model Results

As shown in Table 2, the outer loadings for all items were 0.721 or higher and statistically significant (*p* < 0.001), with the exception of the third item related to technical competencies, which had an outer loading of 0.663. Furthermore, Table 3 illustrates that the composite reliabilities (CRs) of the constructs varied between 0.859 and 0.956, thereby affirming both item and construct reliabilities. The average variance extracted (AVE) values were 0.580 or greater, supporting convergent validity ([13]). Discriminant validity was established as the square root of each construct’s AVE exceeded the correlations between that construct and others ([11]).

Table 4 reveals that online learning readiness had a significant correlation with online self-regulated English learning (*r* = 0.575, *p* < 0.001), satisfaction with the online English learning experience (*r* = 0.536, *p* < 0.001), and perceived English self-efficacy (*r* = 0.633, *p* < 0.001). Additionally, online self-regulated English learning was significantly correlated with satisfaction with the online English learning experience (*r* = 0.461, *p* < 0.001) and perceived English self-efficacy (*r* = 0.477, *p* < 0.001). Moreover, perceived English self-efficacy was significantly associated with satisfaction with the online English learning experience (*r* = 0.464, *p* < 0.001).

### 4.2. Higher-Order Constructs and Inner Model Results

The results of PLS-SEM indicated that online learning readiness, online self-regulated English learning, and perceived English self-efficacy could be conceptualized as second-order constructs. As illustrated in Figure 2, the standardized path coefficients for the three second-order constructs and the inner model are presented. The second-order online learning readiness comprises three key factors. The path coefficients indicated that social competencies with instructors emerged as the most important factor (*β* = 0.917, *p* < 0.001), followed by social competencies with classmates (*β* = 0.828, *p* < 0.001) and technical competencies (*β* = 0.671, *p* < 0.001). The second-order construct of online self-regulated English learning consists of four factors, with goal setting identified as the most important factor (*β* = 0.886, *p* < 0.001), followed closely by time management (*β* = 0.878, *p* < 0.001), self-evaluation (*β* = 0.817, *p* < 0.001), and environmental structuring (*β* = 0.744, *p* < 0.001). The second-order construct of perceived English self-efficacy includes four factors, where speaking was found to be the most important factor (*β* = 0.951, *p* < 0.001), followed by reading (*β* = 0.912, *p* < 0.001), writing (*β* = 0.912, *p* < 0.001), and listening (*β* = 0.895, *p* < 0.001).

Figure 2 illustrates that online learning readiness significantly influenced satisfaction with the online English learning experience (*β* = 0.333, *p* < 0.001), thereby confirming H1. Additionally, online learning readiness was found to significantly affect online self-regulated English learning (*β* = 0.583, *p* < 0.001), while online self-regulated English learning also had a significant effect on satisfaction with the online English learning experience (*β* = 0.197, *p* < 0.05), supporting H2 and H3. Furthermore, the findings indicate that online self-regulated English learning partially mediated the relationship between online learning readiness and satisfaction with the online English learning experience, which supported H4. However, online self-regulated English learning had a weak and insignificant effect on perceived English self-efficacy (*β* = 0.165, *p* > 0.05), leading to the rejection of H5. In contrast, online learning readiness significantly influenced perceived English self-efficacy (*β* = 0.538, *p* < 0.001), thus supporting H6. Perceived English self-efficacy, however, had a weak and insignificant effect on satisfaction with the online English learning experience (*β* = 0.171, *p* > 0.05), resulting in the rejection of H7 and H8.

Figure 2 also reveals that the model could be used to predict students’ satisfaction with the online English learning experience. The coefficient of determination (*R*^2^) for satisfaction with the online English learning experience was found to be 0.36, suggesting that 36 percent of the variance in satisfaction with the online English learning experience was attributed to changes in online learning readiness, both directly and indirectly through online self-regulated English learning.

### 4.3. Mediation Analysis Using Hayes’ PROCESS Macro

To validate the mediating roles of online self-regulated English learning and perceived English self-efficacy in the relationship between online learning readiness and satisfaction with the online English learning experience, latent variable scores were extracted from SmartPLS and entered into an IBM SPSS 26.0 data file. A multiple regression analysis was performed, treating online learning readiness, online self-regulated English learning, and perceived English self-efficacy as independent variables, while satisfaction with the online English learning experience served as the dependent variable. The variance inflation factors (VIFs) for the independent variables were 2.015, 1.562, and 1.725, all of which are below the 3.0 threshold, suggesting that multi-collinearity was not a concern ([13]). Model 6 of Hayes’ PROCESS Macro was utilized to assess the significance of both direct and indirect effects among the variables under study. A total of five thousand bootstrap samples were used to generate 95% bias-corrected confidence intervals. The results of the mediation analysis are presented in Table 5. The findings confirmed that online learning readiness had a direct influence on satisfaction with the online English learning experience (*b* = 0.333, *SE* = 0.090, *p* < 0.001) and an indirect effect through self-regulated English learning (*b* = 0.115, *SE* = 0.054, *p* < 0.05). Furthermore, the indirect effect of online learning readiness on satisfaction with the online English learning experience via perceived English self-efficacy was not significant (*b* = 0.092, *SE* = 0.068, *p* > 0.05, 95%CI [−0.027, 0.241]).

## 5. Discussion

This cross-sectional study indicates that respondents perceived themselves as having a high level of technical competencies in online learning, as evidenced by mean scores ranging from 3.47 to 3.79 across four items. This finding is expected, given that this cohort of students, referred to as Gen Z, has been exposed to the Internet and social media from an early age, which has fostered their comfort in surfing the Internet and completing tasks online. Conversely, these Chinese students acknowledged their deficiencies in general listening and reading skills, as reflected in the mean scores of two items—“I can understand English movies (without Chinese subtitles)” and “I can read English newspapers”—which were recorded at only 2.72, the lowest among all measurement items.

This study also seeks to empirically evaluate the proposed hypotheses regarding the impact of online learning readiness and online self-regulated English learning on satisfaction with the online English learning experience among the undergraduate students during the COVID-19 pandemic. The analysis of higher-order constructs indicates that students’ overall online learning readiness was significantly dependent on their social competencies with instructors, with a standardized path coefficient of 0.917, followed by their social competencies with classmates, with a standardized path coefficient of 0.828 (Figure 2). Consequently, it is essential for instructors to devise strategies that foster engagement with students and facilitate interactions among students in online settings. Additionally, prior research has highlighted the critical role of social presence in achieving effective e-learning outcomes ([47]) and has shown that East Asian EFL students tend to rely more on their teachers, often avoiding direct participation in the learning process ([6]; [27]). This tendency is particularly evident among the student participants in this study, predominantly freshmen (59.5%), who are navigating the transition from secondary to tertiary education. The PLS-SEM findings validated the mediating role of self-regulated English learning in the relationship between online learning readiness and students’ satisfaction with their online English learning experience. Furthermore, the results demonstrated that students’ online learning readiness significantly and substantially impacted their perceived English self-efficacy, with a standardized path coefficient of 0.583 (*p* < 0.001). However, online self-regulated English learning did not significantly affect perceived English self-efficacy, nor did perceived English self-efficacy have a significant influence on students’ satisfaction with the online English learning experience.

The mediation analysis conducted in this study confirmed that self-regulated English learning played a mediating role in the relationship between online learning readiness and satisfaction with the online English learning experience. This finding aligns with previous research indicating a positive correlation between students’ online learning readiness and their satisfaction with the online English learning experience ([42]), as well as the positive association between students’ online learning satisfaction and their online self-regulated English learning ([5]; [21]). While the current study did not find a significant relationship between online self-regulated English learning and perceived English self-efficacy, it is important to highlight that students hold certain kinds of attitudes in believing in goal setting as the most crucial factor in managing their online self-regulated English learning. They also regarded speaking skills as the most important aspect of their perceived English self-efficacy, which influenced self-regulated online English. This suggests that students are prepared to embrace the online learning environment, provided their beliefs in self-efficacy and self-regulated strategies are further enhanced. This presents a crucial opportunity for teachers to engage and provide additional support for students’ self-regulated online learning, as those who believe in their ability to utilize self-regulatory skills tend to be more motivated in their learning endeavors ([2]; [23]). Establishing clear learning goals can further enhance students’ feelings of self-efficacy and self-regulation in their online English learning. Additionally, it is imperative for instructors to make every effort to avoid discouraging less-motivated students. The integration of multimedia resources and interactive activities into online platforms, along with the provision of timely and constructive feedback, is essential to assist these students in recognizing their progress and identifying areas for improvement.

### 5.1. Theoretical Implications

This research established a model linking belief, behavior, performance, and satisfaction in the context of online English learning. Specifically, online learning readiness as students’ beliefs about their own technical and social competencies was found to significantly influence students’ online self-regulated English learning behaviors in the forms of goal setting, environmental structuring, time management, and continuous self-evaluation. Both online learning readiness (belief) and online self-regulated English learning (behavior) had effects on students’ perceived English self-efficacy (performance). Furthermore, students’ perceived English self-efficacy (performance), along with online learning readiness (belief) and online self-regulated English learning (behavior), contributed to their satisfaction with the online English learning experience. This proposed model is anticipated to be applicable in explaining the adoption of online technology learning such as ChatGPT and other large language models and various other online learning formats.

### 5.2. Practical Implications

This research advances our understanding of the effects of students’ online learning readiness and online self-regulated English learning on their satisfaction with the online English learning experience. The various levels of factors identified in this research also provide valuable insights for the design and implementation of effective online self-regulated learning. Consequently, this research widens and extends the English learning literature.

From a practical standpoint, it is essential for instructors to actively support students, even in the absence of face-to-face instruction during the online self-regulated learning process. For example, instructors should monitor students’ learning activities on the online platform and offer prompt assistance as needed. In addition to instructor support, incorporating group discussion activities on various online platforms is highly advisable, such as breakout rooms in ZOOM or collaborative discussions in CANVAS. This approach fosters an online learning community that encourages student interaction and collaborative learning, which can help alleviate online learning anxiety. Furthermore, since online self-regulated English learning is crucial in mediating the relationship between online learning readiness and satisfaction with the online English learning experience, it is vital to equip students with effective self-regulatory strategies for their online studies. Instructors can design and implement specific lesson plans and integrated frameworks that encourage students to set goals, manage their time efficiently, and utilize smart learning strategies, including the use of generative artificial intelligence technologies to enhance their self-regulatory capabilities. The positive correlations observed between perceived English self-efficacy and other constructs, such as online learning readiness, self-regulated English learning, and satisfaction with the online English learning experience, further underscore the necessity for English instructors to encourage students to acquire and utilize diverse self-efficacy skills on e-learning platforms. Although perceived English self-efficacy did not demonstrate a significant direct effect on satisfaction with the online English learning experience, an indirect relationship may be present. Consequently, the findings of this research serve as a valuable reference for researchers seeking to investigate the factors that can enhance students’ self-regulation behavior and satisfaction within the online learning environment.

All in all, this research is particularly relevant in our increasingly technology-driven society, where students can access a wide array of online learning resources. Gaining a comprehensive understanding of students’ self-regulated and fulfilling English learning experience is essential for fostering an autonomous learning mindset in online environments.

### 5.3. Limitations and Future Research

This research presents several limitations. Firstly, it was a cross-sectional study that required Macau’s university business students to reflect on their online English learning experiences during the COVID-19 pandemic. Given that the online English learning period lasted only three weeks in August 2022 and all participants were undergraduate business students, caution is advised when generalizing the findings. Future research could investigate whether students from different faculties, programs, and universities hold different views on the online English learning experience. Secondly, the data collected were self-reported data, which may introduce common method bias ([31]). A post hoc analysis using Harman’s single-factor test was conducted ([14]). The results revealed that no single factor dominated and the first unrotated factor accounted for only 40.5% of the total variance, suggesting that common method variance is unlikely to be a significant concern in this research. Thirdly, the hypotheses were formulated based on existing literature, and a cross-sectional study design was employed to explore the impact of online learning readiness on satisfaction with the online English learning experience directly and indirectly through self-regulated English learning and perceived English self-efficacy. Future research should consider employing longitudinal or experimental research designs to further confirm their causal relationships. Fourthly, our research concentrated on students who are required to learn English as a second language due to its importance for their future careers. As a result, most students exhibit a greater motivation to participate in online English learning, even amidst the challenges posed by the COVID-19 pandemic. Future studies could examine the model’s relevance to other language courses that students opt for as electives to fulfill credit requirements or to fill up their schedules. Lastly, this research did not examine instructors’ teaching approaches in online environments. Future research could assess how instructors’ teaching approaches and perceived instructor support may directly or indirectly affect students’ online self-regulated English learning and their satisfaction with the online English learning experience. Past research has indicated that perceived supervisor support significantly influences students’ satisfaction with their internship experience among Chinese students ([39]).

## 6. Conclusions

This study explored two key questions regarding the relationships between students’ online learning readiness, online self-regulated English learning, perceived English self-efficacy, and satisfaction with the online English learning experience. The findings revealed that online learning readiness had a significant direct and indirect impact on students’ satisfaction with their online English learning experience, mediated by their self-regulated English learning. Additionally, online learning readiness was found to affect students’ perceived English self-efficacy. However, online self-regulated English learning did not significantly affect students’ perceived English self-efficacy, nor did perceived English self-efficacy have a significant relationship with satisfaction with the online English learning experience.

## Figures and Tables

**Figure 1 behavsci-15-00093-f001:**
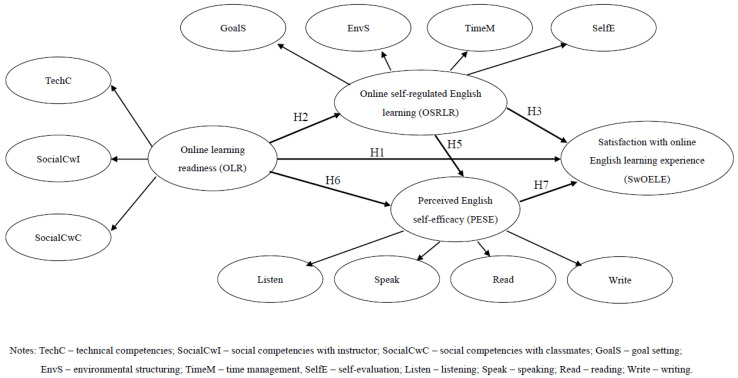
Theoretical model of this study.

**Figure 2 behavsci-15-00093-f002:**
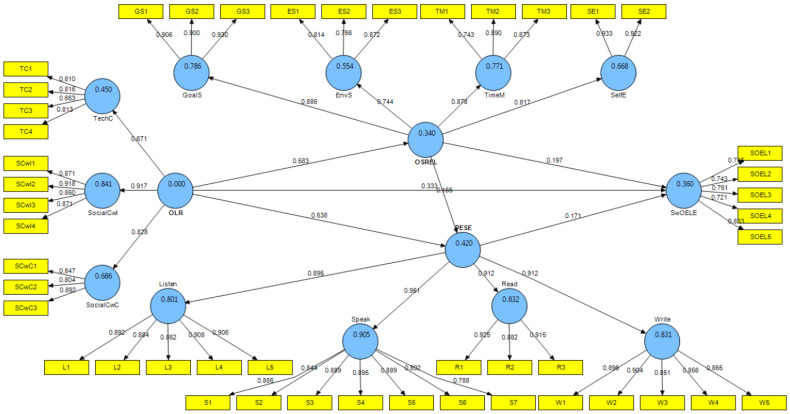
PLS-SEM results.

**Table 1 behavsci-15-00093-t001:** Demographic profile of respondents (*N* = 163).

Variable	Class	Frequency	%
Gender	Male	78	47.9
	Female	85	52.1
Age (years)	<20	71	43.6
	20 to <22	52	31.8
	22 to <24	14	8.6
	24 to <26	7	4.3
	26 or above	19	11.7
Year of study	1	97	59.5
	2	10	6.1
	3	26	16.0
	4	30	18.4
Specialization/program	Management	110	67.5
	e-Commerce	18	11.0
	Accounting	34	21.5
Preferred mode of English learning	Face-to-face only	75	46.0
	Hybrid	60	36.8
	Online only	28	17.2

**Table 2 behavsci-15-00093-t002:** Constructs and measurement items.

Constructs and Measurement Items	Mean (SD)	Outer Loadings	Alpha
Online learning readiness—technical competencies (TechC)			0.784
- I have a sense of self confidence in using online technologies for specific tasks.	3.55 (0.911)	0.810	
- I am proficient in using a wide variety of online technologies.	3.47 (0.918)	0.816	
- I feel comfortable surfing the Internet.	3.79 (1.045)	0.663	
- I am competent at integrating online technologies into my learning activities.	3.50 (0.945)	0.813	
Online learning readiness—social competencies with instructor during the online English course (SocialCwI)			0.903
- I am confident that I can clearly ask my instructor questions.	3.20 (1.030)	0.871	
- I am confident that I can initiate discussions with the instructor.	3.05 (1.070)	0.918	
- I am confident that I can seek help from the instructor when needed.	3.28 (1.051)	0.860	
- I am confident that I can timely inform the instructor when unexpected situations arise.	3.13 (1.128)	0.871	
Online learning readiness—social competencies with classmates during the online English course (SocialCwC)			0.806
- I am confident that I can initiate social interaction with classmates.	3.12 (1.097)	0.847	
- I am confident that I can pay attention to other students’ social actions.	3.28 (0.970)	0.804	
- I am confident that I can apply different social interaction skills depending on situations.	3.19 (1.069)	0.892	
Online self-regulated English learning—goal setting (GoalS)			0.899
- I set short-term (daily) and long-term (monthly) goals when learning English online.	2.78 (1.111)	0.906	
- I set standards for my assignments when learning English online.	3.08 (1.042)	0.900	
- I set goals to help me manage study time for my online English learning.	2.95 (1.110)	0.930	
Online self-regulated English learning—environmental structuring (EnvS)			0.754
- I choose a good location for learning English online to avoid too many distractions.	3.47 (1.073)	0.814	
- I know where I can learn English online most efficiency.	3.40 (1.080)	0.766	
- I choose a time with few distractions when studying English online.	3.44 (1.043)	0.872	
Online self-regulated English learning—time management (TimeM)			0.786
- I make use of my fragmental time to learn English online.	3.09 (1.116)	0.743	
- I try to schedule the same time every day to learn English online and I observe the schedule.	2.87 (1.101)	0.890	
- Although we do not have to attend daily online English classes, I still try to distribute my studying time evenly across days.	2.96 (1.062)	0.873	
Online self-regulated English learning—self-evaluation (SelfE)			0.838
- I summarize my online English learning to examine my understanding of what I have learnt.	3.03 (1.009)	0.933	
- I communicate with my instructor to find out how I am doing with my online English learning.	2.80 (1.154)	0.922	
Perceived English self-efficacy—listening (Listen)			0.935
- I can understand stories told in English.	3.17 (1.028)	0.892	
- I can understand English lectures of general topics.	3.07 (1.106)	0.884	
- I can understand English songs (without Chinese subtitles).	2.92 (1.094)	0.862	
- I can understand English TV/online programs (without Chinese subtitles).	2.75 (1.095)	0.908	
- I can understand English movies (without Chinese subtitles).	2.72 (1.108)	0.906	
Perceived English self-efficacy—speaking (Speak)			0.946
- I can describe my university to other people in English.	3.01 (1.063)	0.886	
- I can describe the way to the University from the place I live in English.	3.10 (1.098)	0.844	
- I can tell a story in English.	2.94 (1.101)	0.889	
- I can ask my English instructor questions in English.	3.29 (1.153)	0.895	
- I can answer my English instructor’s questions in English.	3.29 (1.065)	0.889	
- I can introduce myself in English (without reading notes).	3.44 (1.083)	0.892	
- I can do English presentations in class (without reading notes).	2.75 (1.139)	0.788	
Perceived English self-efficacy—reading (Read)			0.893
- I can understand online English news.	2.78 (1.094)	0.925	
- I can read short English narrative.	3.24 (1.099)	0.882	
- I can read English newspapers.	2.74 (1.174)	0.915	
Perceived English self-efficacy—writing (Write)			0.923
- I can write coherent English sentences.	3.26 (1.148)	0.896	
- I can leave a note for another student in English.	3.25 (1.135)	0.904	
- I can write English compositions assigned by my English instructor.	3.45 (1.128)	0.851	
- I can punctuate correctly when I write English essays.	3.29 (1.081)	0.856	
- I can use accurate grammar when I write English essays.	2.89 (1.160)	0.865	
Satisfaction with online English learning experience (SwOELE)			0.820
- I am satisfied with the chance to interact more with English instructor online.	3.37 (1.048)	0.745	
- I hardly listened to English classes in face-to-face classroom environment but I can focus more on online English learning.	2.78 (1.117)	0.743	
- Comparing to face-to-face classroom environment, I was less distracted in online English learning environment.	2.83 (1.153)	0.761	
- Online English learning is time-flexible.	3.58 (1.154)	0.721	
- Online English learning allows more autonomy and I can learn English in my own way.	3.48 (1.062)	0.883	

**Table 3 behavsci-15-00093-t003:** CRs, AVEs, and correlations between constructs.

	CR	AVE	TechC	SocialCwl	SocialCwC	GoalS	EnvS	TimeM	SelfE	Listen	Speak	Read	Write	SwOELE
TechC	0.859	0.606	** *0.778* **											
SocialCwl	0.932	0.775	0.434	** *0.880* **										
SocialCWC	0.885	0.720	0.330	0.656	** *0.849* **									
GoalS	0.937	0.833	0.277	0.446	0.510	** *0.913* **								
EnvS	0.859	0.670	0.325	0.314	0.454	0.505	** *0.819* **							
TimeM	0.875	0.702	0.185	0.387	0.488	0.703	0.557	** *0.838* **						
SelfE	0.925	0.860	0.166	0.548	0.559	0.652	0.474	0.638	** *0.927* **					
Listen	0.950	0.793	0.263	0.628	0.424	0.422	0.339	0.364	0.511	** *0.891* **				
Speak	0.956	0.757	0.281	0.661	0.469	0.365	0.334	0.373	0.427	0.784	** *0.879* **			
Read	0.934	0.824	0.213	0.548	0.404	0.391	0.259	0.349	0.363	0.802	0.815	** *0.908* **		
Write	0.942	0.765	0.350	0.542	0.428	0.345	0.309	0.313	0.360	0.709	0.829	0.808	** *0.875* **	
SwOELE	0.873	0.580	0.469	0.423	0.479	0.415	0.434	0.346	0.345	0.386	0.422	0.435	0.475	** *0.762* **

Note: The bold italicized numbers in the diagonal are the square roots of AVEs.

**Table 4 behavsci-15-00093-t004:** Correlations between higher-order constructs and SwOELE.

	OLR	OSREL	PESE	SwOELE
Online learning readiness (OLR)	1.000			
Online self-regulated English learning (OSREL)	0.575	1.000		
Perceived English self-efficacy (PESE)	0.633	0.477	1.000	
SwOELE	0.536	0.461	0.464	1.000

**Table 5 behavsci-15-00093-t005:** Results of mediation analysis.

Relationship	Effect (*b*)	*SE*	*t*	*p*	LLCI	ULCI
Direct effect of X (ORL) on Y (SwOELE)	0.333	0.090	3.694	0.0003	0.155	0.511
Indirect effects of X (ORL) on Y (SwOELE)	Effect (*b*)	Boot*SE*			BootLLCI	BootULCI
Total	0.224	0.089			0.059	0.414
Ind1: OLR → SREL → SwOELE	0.115	0.054			0.017	0.227
Ind2: OLR → PESE → SwOELE	0.092	0.068			−0.027	0.241
Ind3: OLR → SREL → PESE → SwOELE	0.017	0.016			−0.008	0.053

## Data Availability

The data presented in this study are available on request from the corresponding author due to privacy reasons.

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
