# Peer review of "Effects of Online Learning Readiness and Online Self-Regulated English Learning on Satisfaction with Online English Learning Experience During the COVID-19 Pandemic"

_behavsci, 2025, doi:10.3390/bs15010093_

Round 1

Reviewer 1 Report

Comments and Suggestions for Authors

It is not often I get to review such a well written article in excellent English, my congratulations to the authors.  I have a few comments.

1. Could you add a couple of sentences explaining the different statistical analyses, e.g. explaining what you are analysing, prior to the analyses?

2. A couple of extra sentences explaining the findings would be appreciated.

3. I would draw your attention to 5. Discussion. This subsection does not hold the same level of quality as the rest of the article, there are unclear sentences  and even errors. E.g. page 11, lines365-367 of an example. This part needs to be presented more clearly.

4. I greatly appreciated 5.2. Practical Implications. This is, in my opinion,  the most important part of the article. I strongly suggest going into further detail about how to equip students for self-regulated learning in particular.

Author Response

General comment: It is not often I get to review such a well written article in excellent English, my congratulations to the authors.  I have a few comments.

Our response: Thanks so much for your comment. We have studied your comments thoroughly and have made the necessary changes to the manuscript.

Comment 1: Could you add a couple of sentences explaining the different statistical analyses, e.g. explaining what you are analysing, prior to the analyses?

Response 1: Thanks very much for your suggestion. In the revised manuscript, we have added some sentences in Section 3.3. Data Analysis Approach to clarity the aims of the various statistical analyses prior to their execution. Consequentially, the updated section is:

“              Data were entered into an Excel file, where means and standard deviations were calculated to evaluate the agreement levels with the Likert scale items. This analysis provided insights into respondents’ perceptions of their readiness for online learning, the degree to which they engaged in online self-regulated English learning, their self-efficacy in English across listening, speaking, reading, and writing, as well as their satisfaction with the online English learning experience and the associated variances. To explore the relationships among constructs and the loadings of indicators, partial least squares-structural equation modeling (PLS-SEM) was utilized. PLS-SEM is particularly advantageous for small sample sizes and complex models, especially when predicting a specific outcome such as students’ satisfaction with their online English learning experience [38]. The analysis was conducted using SmartPLS, which was obtained from Ringle et al. [39]. SmartPLS generates outer loadings for all measurement items, along with Cronbach’s alpha values, composite reliabilities (CRs), and average variance extracted (AVE) values. These metrics were utilized to evaluate item and construct reliabilities, as well as convergent and discriminant validities [38,39]. The standardized path coefficients for the inner model were determined to reveal the direction, magnitude, and significance of the structural relationships among the primary constructs, which include online learning readiness, online self-regulated English learning, perceived English self-efficacy, and satisfaction with the online English learning experience. Furthermore, the PROCESS Macro [40] was employed to validate the mediating roles of self-regulated English learning and perceived English self-efficacy in the relationship between online learning readiness and satisfaction with the online English learning experience.”

Comment 2: A couple of extra sentences explaining the findings would be appreciated.

Response 2: Thanks very much for your suggestion. In the revised manuscript, we have included an additional paragraph at the beginning of Section 5. Discussion to emphasize several noteworthy observations and findings.

“5. Discussion

This cross-sectional study indicates that respondents perceived to have a high level of technical competencies in online learning, as evidenced by mean scores ranging from 3.47 to 3.79 across four items. This finding is expected, given that this cohort of students, referred to as Gen Z, has been exposed to the Internet and social media from an early age, which has fostered their comfort in surfing the Internet and completing tasks online. Conversely, these Chinese students acknowledged their deficiencies in general listening and reading skills, as reflected in the mean scores of two items – “I can understand English movies (without Chinese subtitles)” and “I can read English newspapers” – which were recorded at only 2.72, the lowest among all measurement items.

     The study also seeks to empirically…”

Comment 3: I would draw your attention to 5. Discussion. This subsection does not hold the same level of quality as the rest of the article, there are unclear sentences and even errors. E.g. page 11, lines365-367 of an example. This part needs to be presented more clearly.

Response 3: Thanks very much for your comment. The sentences found in lines 365-367 of the original manuscript – “As shown in the higher-order constructs and inner model results, students concerned highly on their own social competencies with instructors and classmates while engaging in online learning” – have been revised to:

“…The analysis of higher-order constructs indicates that students’ overall online learning readiness was dependent significantly on their social competencies with instructors, with a standardized path coefficient of 0.917, followed by their social competencies with classmates, with a standardized path coefficient of 0.828 (Figure 2). Consequently, it is essential for instructors to devise strategies that foster engagement with students and facilitate interactions among students in online settings. Additionally, prior research …”

Furthermore, this particular paragraph has been restructured to provide a more comprehensive presentation and discussion of the PLS-SEM findings in the revised manuscript.

Comment 4: I greatly appreciated 5.2. Practical Implications. This is, in my opinion, the most important part of the article. I strongly suggest going into further detail about how to equip students for self-regulated learning in particular.

Response 4: Thanks very much for your comment and suggestion. In the revised manuscript, we have rewritten and elaborated on the second paragraph as follows:

“              From a practical standpoint, it is essential for instructors to actively support students, even in the absence of face-to-face instruction during the online self-regulated learning process. For example, instructors should monitor students’ learning activities on the online platform and offer prompt assistance as needed. In addition to instructor support, incorporating group discussion activities on various online platforms is highly advisable, such as breakout rooms in ZOOM or collaborative discussions in CANVAS. This approach fosters an online learning community that encourages student interaction and collaborative learning, which can help alleviate online learning anxiety. Furthermore, since online self-regulated English learning is crucial in mediating the relationship between online learning readiness and satisfaction with the online English learning experience, it is vital to equip students with effective self-regulatory strategies for their online studies. Instructors can design and implement specific lesson plans and integrated frameworks that encourage students to set goals, manage their time efficiently, and utilize smart learning strategies, including the use of generative artificial intelligence technologies to enhance their self-regulatory capabilities. The positive correlations observed be-tween perceived English self-efficacy and other constructs, such as online learning readiness, self-regulated English learning, and satisfaction with the online English learning experience further underscore the necessity for English instructors to encourage students to acquire and utilize diverse self-efficacy skills on e-learning platforms. Although perceived English self-efficacy did not demonstrate a significant direct effect on satisfaction with the online English learning experience, an indirect relationship may be present. Consequently, the findings of this research serve as a valuable reference for researchers seeking to investigate the factors that can enhance students’ self-regulation behavior and satisfaction within the online learning environment.”

Reviewer 2 Report

Comments and Suggestions for Authors

This is an interesting paper.

It should clarify how ethical approval was obtained for the research, e.g., how learners were informed of the study, and how they consented to participate.

There are too many hypotheses in this paper. I am uncomfortable with using a data sample of 163 students to test 8 hypotheses. I would have thought 2-3 hypotheses would be far more appropriate. I feel this could benefit from a rethink.

The work is based on statistical analysis, and causal relationships are assumed based on statistical relationships. This should be acknowledged as a weakness of the work. The stats might suggest relationships, but they do not prove their existence.

There is a lot of hypothesis testing. The work needs to clarify how multiple comparison issues are addressed or explain the rationale for not considering them.

There is some overstating of the significance of the work, e.g., lines 410-411. I suggest deleting significant on line 411 and significant on line 426.

Author Response

Comment 1: This is an interesting paper.

Response 1: Thanks very much for your comment. We have carefully considered your comments and made the necessary revisions to the manuscript. We sincerely hope that the changes made adequately address your concerns in the revised version.

Comment 2: It should clarify how ethical approval was obtained for the research, e.g., how learners were informed of the study, and how they consented to participate.

Response 2: Thanks very much for your comment. As stated in the Institutional Review Board Statement on page 13, the ethical review and approval for this study were waived in accordance with the local legislation and institutional requirements (Article 32 of Measures for Ethical Review of Life Sciences and Medical Research Involving Human Beings of China; further details are available at https://www.gov.cn/zhengce/zhengceku/2023-02/28/content_5743658.htm , accessed on 15 November 2024), since this research did not involve clinical trials or manipulations involving humans or animals. Regarding how leaners were informed of the study, and how they consented to participate, we have provided clarification in section 3.1 as follows:

“…This study involved undergraduate business students enrolled in English courses at a public university in Macau, who were invited to participate in a questionnaire survey conducted from November 2022 to March 2023, during and immediately following the COVID-19 pandemic. The research team collaborated with English instructors to distribute the questionnaires to 250 students during their classes. … Participation in the study was entirely voluntary and anonymous. The questionnaire commenced with a brief overview of the study’s purpose. Students were then required to provide their consent before proceeding with the questionnaire. Assurance was given regarding the confidentiality of their responses, and participants had the option to withdraw from the survey at any point. Due to an outbreak of the COVID-19 omicron variant in Macao from 18 June 2022 to 2 August 2022 [37], universities mandated that instructors conducted their first few lectures online in August 2022. Face-to-face classes were only allowed to resume after 1 September 2022. When completing the questionnaire, all participants were asked to reflect on their online English learning experience in August 2022.

Comment 3: There are too many hypotheses in this paper. I am uncomfortable with using a data sample of 163 students to test 8 hypotheses. I would have thought 2-3 hypotheses would be far more appropriate. I feel this could benefit from a rethink.

Response 3: Thanks very much for your comment. We reached out to 250 students via their English instructors; however, only 163 completed the questionnaire. Despite this, PLS-SEM is an effective method for analyzing responses from small sample sizes, such as our case, and complex models. Consequently, all measurement items were validated through the outer models, and the eight hypotheses were rigorously tested using the inner model (see Section 4. Results). To confirm that the study possessed sufficient power to identify significant effects, we conducted a priori power analysis utilizing G*Power 3.1 software (see Section 3.1). By inputting our model parameters, we determined that the minimum sample size required to identify a statistically significant effect was 77, while our final sample consisted of 163 participants.

Comment 4: The work is based on statistical analysis, and causal relationships are assumed based on statistical relationships. This should be acknowledged as a weakness of the work. The stats might suggest relationships, but they do not prove their existence.

Response 4: Thanks very much for your comment. As suggested by you, we have added the following sentences in the limitations:

“…Thirdly, the hypotheses were formulated based on existing literature, and a cross-sectional study design was employed to explore the impact of online learning readiness on satisfaction with the online English learning experience directly and indirectly through self-regulated English learning and perceived English self-efficacy. Future research should consider employing longitudinal or experimental research designs to further confirm their causal relationships….”

Comment 5: There is a lot of hypothesis testing. The work needs to clarify how multiple comparison issues are addressed or explain the rationale for not considering them.

Response 5: Thanks very much for your comment. We first tested our hypotheses using PLS-SEM. After that, we validated the mediating roles of self-regulated English learning and perceived English self-efficacy using Hayes’ PROCESS Macro.

Comment 6: There is some overstating of the significance of the work, e.g., lines 410-411. I suggest deleting significant on line 411 and significant on line 426.

Response 6: Thanks very much for your comment. In the revised manuscript, we have deleted the word - “significant” in these sentences.